# Age of Bilingual Onset Shapes the Dynamics of Functional Connectivity and Laterality in the Resting-State

**DOI:** 10.3390/brainsci13091231

**Published:** 2023-08-23

**Authors:** Yucen Sheng, Songyu Yang, Juan Rao, Qin Zhang, Jialong Li, Dianjian Wang, Weihao Zheng

**Affiliations:** 1School of Foreign Languages, Lanzhou Jiaotong University, Lanzhou 730070, China; 2Gansu Provincial Key Laboratory of Wearable Computing, School of Information Science and Engineering, Lanzhou University, Lanzhou 730000, China; 3School of Electronic and Information Engineering, Lanzhou Jiaotong University, Lanzhou 730070, China

**Keywords:** bilingualism, L2 acquisition age, fMRI, dynamic functional connectivity, dynamic laterality

## Abstract

Bilingualism is known to enhance cognitive function and flexibility of the brain. However, it is not clear how bilingual experience affects the time-varying functional network and whether these changes depend on the age of bilingual onset. This study intended to investigate the bilingual-related dynamic functional connectivity (dFC) based on the resting-state functional magnetic resonance images, including 23 early bilinguals (EBs), 30 late bilinguals (LBs), and 31 English monolinguals. The analysis identified two dFC states, and LBs showed more transitions between these states than monolinguals. Moreover, more frequent left–right switches were found in functional laterality in prefrontal, lateral temporal, lateral occipital, and inferior parietal cortices in EBs compared with LB and monolingual cohorts, and the laterality changes in the anterior superior temporal cortex were negatively correlated with L2 proficiency. These findings highlight how the age of L2 acquisition affects cortico-cortical dFC pattern and provide insight into the neural mechanisms of bilingualism.

## 1. Introduction

Bilingualism has been found to bring many cognitive benefits [1], including improved executive function [2], auditory attention [3], and working memory [4]. It may also delay the onset of Alzheimer’s disease by 4–5 years compared with monolinguals [5,6].

Examining how bilingual acquisition affects the brain functional network is helpful for understanding its neural mechanisms, which can also contribute to preventing cognitive decline. With the advancement of neuroimaging technology, studies have progressively reported that bilinguals alter volume and activity relative to monolinguals in certain brain regions associated with executive control, memory, and language processing, such as the left temporal [7], right inferior frontal, and right superior parietal cortices [8]. In addition, the early learning of a second language (L2) can modulate the development of cognitive functions (e.g., stronger inhibitory control [9], higher language proficiency, and better cognitive control [10]) as well as the brain structure (e.g., broader area of the right parietal suprachiasmatic cortex [11]). However, the effects of learning a L2 and its acquisition age on the dynamics of brain functional networks—a crucial question for understanding the neural basis of bilingualism—remain unclear.

Functional magnetic resonance imaging (fMRI) technology has been utilized to investigate the impact of bilingualism on brain activation and functional connectivity (FC). Some previous studies have designed diverse tasks to explore the neural correlates of executive control in bilinguals. The findings revealed that bilinguals demonstrated similar activation of brain regions at both word and sentence levels during task performance regardless of whether participants were using their first or L2 [12,13]. Green et al. found that the left inferior frontal gyrus and anterior cingulate cortex were activated in bilinguals during language switching by using a color–shape switching task [14]. In addition, bilingual individuals also showed greater activation in the left superior temporal gyrus, bilateral cingulate gyrus, right middle frontal gyrus, and right caudate nucleus during Simon’s paradigm task (a test for exploring human cognitive traits) compared with the monolinguals [15]. The alterations in functional activity may also cause FC changes in the resting state of the bilingual brain. For instance, compared with monolinguals, bilinguals demonstrated increased FC within the default mode [16,17], language, and cognitive control networks [18,19], as well as between perceptual/motor regions [20], while also showing decreased FC connecting the dorsal anterior cingulate cortex to the left superior temporal cortex and rolandic operculum, which are key regions for speech and auditory processing [21].

In the neuroimaging studies, FC is an important indicator measuring the coactivation pattern between distributed brain regions. Nevertheless, these studies only provide a static picture of bilingual brain function, assuming that FC remains stable throughout the whole scan period. As the human brain is a dynamic system that continuously reconfigures its functional interactive patterns over time [22], it is necessary to explore bilingualism-related changes using a dynamic brain connection model. The FC of the brain at rest naturally varies over time in both strength and directionality due to fluctuations in the blood-oxygen-level-dependent (BOLD) signal [22]. The sliding window approach is currently the most widely used strategy for investigating these temporal variations in resting-state FC [23]. This method involves segmenting a long time sequence into a series of continuous time periods by shifting a fixed-length time window and then using time points within the window to compute FC. This method enables the quantification of time-varying FC, also known as dynamic functional connectivity (dFC) [22], over the entire scan duration. Indeed, changes in dFC have been reported in multiple neural and physiological processes (e.g., alertness and attention [24]), as well as in brain disorders (e.g., autism spectral disorder, Alzheimer’s disease, and major depressive disorder), such as altered frequency of transitions between different dFC states and state-dependent connectivity changes of the patients [25,26,27,28,29]. Notably, a recent study conducted on the bilingual population of Cantonese–Mandarin has revealed that the age of L2 acquisition affects the temporal neural activation and dynamic topological properties of the language network [30]. These findings suggested that dFC may provide critical information that cannot be captured by conventional static FC analysis and may, therefore, be useful in characterizing the dynamic nature of bilingual brains. Though important, it is still an open question how dFC is altered in English–Spanish bilingual brains and whether these changes occur in a similar manner as in Cantonese–Mandarin speakers.

Our study aims to test the hypothesis that the acquisition age of L2 modulates the dynamics of functional organization in bilingual brains. We analyzed the resting-state fMRI data from 53 individuals who are bilingual in English–Spanish, with varying age of L2 acquisition, as well as data from 31 English monolinguals. The sliding window approach was used to compute the dFC. The comparisons in the temporal properties of connectivity states, state-dependent FC, and activity laterality across time windows in bilingual individuals, as well as their relationships with L1/L2 performance, were conducted to support the hypothesis that the age of L2 onset is associated with changes in dFC throughout the brain.

## 2. Materials and Methods

### 2.1. Participants

Imaging data were downloaded from the OpenNeuro database accessed on 3 April 2022 (https://openneuro.org/datasets/ds001747). This dataset was chosen because of its well-designed data collection, e.g., categorization of three groups based on the L2 acquisition age, and strictly controlled for age, sex, and education (over 80% participants were full-time college students), which can minimize the influences of these factors on statistical analysis. A total of 102 participants aged between 18 and 29 years and had specific language background were recruited at Brigham Young University [17]. Recruitment focused on three types of populations, namely early bilinguals (EBs), late bilinguals (LBs), and monolingual controls (MCs). Individuals were invited to participate based on their survey responses, specifically whether they would fit into one of the three research groups. The EB group was composed of individuals proficient in English and Spanish with the experience of both languages that began before the age of ten. The LB group consisted of individuals proficient in English and Spanish and who started to learn the L2 (Spanish) after the age of fourteen. The MC group included individuals who only had English experience or had experiences with other languages (in addition to English) that were acquired at any age but was self-rated as “Novice” level [17]. The data exclusion criteria were as follows: left-handed, color-deficient or color-blind, suffered a traumatic brain injury, diagnosed with a psychiatric or neurological disorder, and pregnancy. All participants met the criteria for participating in the MRI scans. In addition, participants with evident head movement during scan were also excluded (see Section 2.3 for details). The demographic information is shown in Table 1. The participants’ English proficiency and Spanish proficiency tests were designed as follows: the study design called for all participants to complete an English proficiency test and any participant with experience with the Spanish language (from taking a semester in junior high to growing up with the language) to take a Spanish proficiency test. The tests were provided by Emmersion Learning, which administered the exams via an online platform, with participants completing them from a home computer. Both proficiency tests used elicited imitation (EI), in which the test taker hears an utterance in the target language and is prompted to repeat the utterance as accurately as possible. If the participant is completely unfamiliar with the language of the utterance they hear, each syllable of the utterance will count toward that limited capacity, reducing their ability to accurately repeat the utterance to only a few syllables. In this way, EI can reliably approximate a learner’s proficiency level by measuring the accuracy of the repetition of utterances of increasing length and complexity [17].

### 2.2. MRI Acquisition

T1-weighted and rs-fMR images were acquired on a 3T Siemens TIM Trio MRI scanner using a 32-channel head coil. T1 images were acquired with the following parameters: TR/TE = 1900/2.26 ms, acquisition matrix = 215 × 256, field of view = 218 × 250 mm, slice thickness = 1.0 mm, voxel size = 1.0 × 1.0 × 1.0 mm, flip angle = 9°, and number of slices = 176. The parameters of rs-fMRI were as follows: TR/TE = 875/43.6 ms, slice thickness = 1.8 mm, acquisition matrix = 100 × 100, flip angle = 55°, number of slices = 72, field of view = 180 × 180 mm, voxel size = 1.8 × 1.8 × 1.8 mm, multi-band factor = 8, and volumes per run = 823 (total scan time = 12 min). EPI scans were oriented parallel to the long axis of the hippocampus.

### 2.3. Preprocessing

The rs-fMRI data were preprocessed by using Data Processing of Resting-State fMRI (DPARSF) [31] toolbox in Statistical Parametric Mapping (SPM12) [32]. Briefly, the main preprocessing steps included the removal of 10 initial volumes, time-slicing correction, head motion realignment, normalization to the Montreal Neurological Institute (MNI) space, regression of nuisance covariates (i.e., white matter, gray matter, and cerebrospinal fluid interference signals), linear detrending, band-pass filtering in the range of 0.01–0.1 Hz, and spatial smoothing using a Gaussian kernel of 6 mm full width at half maximum. To minimize the potential effects of head motion on FC, we excluded the participants with head motion over than 2 mm translation or 2° rotations, resulting in the exclusion of 4 EBs, 2 LBs, and 2 monolinguals.

### 2.4. Dynamic Functional Connectivity

The dFC analysis based on sliding windows was performed by using Dynamic Brain Connectome (DynamicBC) toolbox [33]. We used a tapered window created by convolving a rectangle (width = 30 TRs) according to a previous literature indicating that the correlation between the dynamic laterality indicators and the results of the window length of 30 TR gradually decreased with increases in window length in the range of 30–90 TRs [34]. Sliding the time window with a step of 1 TR along the full-length time series (813 TRs) resulted in 784 consecutive windows across the entire scan. We segmented the cerebral cortex into 360 brain regions according to the multimodal parcellation (MMP) atlas [35], and a 360 × 360 Pearson correlation matrix was calculated for each window. Pearson correlation coefficients were converted to z-values by using Fisher’s z transformation. We categorized the 360 brain regions into twelve functional networks, namely primary visual (Vis1), secondary visual (Vis2), somatomotor (SMN), cingulo-opercular (CON), dorsal attention (DAN), language (LAN), frontoparietal (FPN), auditory (AUD), default mode (DMN), posterior multimodal (PMN), ventral multimodal (VMN), and orbito-affective (ORA) networks, through the Cole-Anticevic approach [36]. The k-means clustering method was then conducted on the functional network matrices of all participants to estimate reoccurring FC states. The number of clusters (K) was initially set to 2–10, and the optimal K was estimated by using the Elbow criterion [37]. We computed the dwell time, the times of state transition, and the fraction time, measuring the number of consecutive windows in a certain state, the times that the FC state switched from one to another, and the proportion of time spent in each state, respectively, to represent the dynamic changes of functional brain networks over time.

### 2.5. Dynamic Laterality

Three laterality indices, including the number of laterality reversal (LR), the dynamic lateral index (DLI), and the laterality fluctuations (LF) of regional time series [34], were used to characterize the lateralization of dynamics between bilateral hemispheres. The DLI at the t-th sliding time window is defined as
$$DLI_{t}=r(ROI_{i},GS_{L})-r(ROI_{i},GS_{R})$$

The ROI_i_ indicates BOLD time series of the i-th brain region; GS_L_ and GS_R_ indicate the global signals of left and right hemispheres, respectively. The LR of a brain region refers to the times of zero-crossing in DLI (left→right/right→left changes) between two continuous time windows, representing the dynamic integration of information across hemispheres. LF refers to the standard deviation of the lateralized time series that corresponds to the magnitude of fluctuations in the laterality of a brain region. The LF of each brain region is defined as
$$LF=\sqrt{\frac{\sum_{i=1}^{n}{\left(x_{i}-\bar{x}\right)}^{2}}{n-1}}$$
x_i_ indicates the i-th subject’s value of laterality time series, and $\bar{x}$ indicates the mean value of the time series. The analytic pipeline is shown in Figure 1.

### 2.6. Statistical Analysis

For demographic information, we used the analysis of variance (ANOVA), and to examine the difference among the three groups on age, years of education, ethnicity, and Spanish and English performance scores, we used the Tukey post hoc test. The chi-square test was performed to examine the sex difference between each pair of the three groups. For brain properties, we firstly used the ANOVA to identify brain regions that exhibited significant group-level differences (*p* < 0.05). The analysis of covariance (ANCOVA), with age, ethnicity, and years of education as covariates, was then performed as a post hoc test to examine the difference between two groups. Multiple comparisons were corrected by using the false discovery rate (FDR) approach at the level of q = 0.05. The association between dynamic laterality metrics and L2 performance was evaluated via Spearman correlation analysis.

## 3. Results

### 3.1. Demographic Analysis

The age of monolinguals significantly differed from the EB and LB groups (ANOVA, *p* < 0.05; post hoc test *p* = 0.006 and 0.031, respectively), and significant differences in ethnicity were found between EB and the other two groups (ANOVA, *p* < 0.05; post hoc test *p*s < 0.0001). The Spanish performance score significantly differed between each pair of groups (ANOVA, *p* < 0.05), i.e., EB vs. LB (post hoc test *p* = 0.009), EB vs. monolinguals (post hoc test *p* < 0.0001), and LB vs. monolinguals (post hoc test *p* < 0.0001). To minimize the influences of age, ethnicity, and years of education on the following analysis, these factors were included as covariates in the ANCOVA. In addition, the three groups did not show statistic differences in sex (chi-square test, *p* > 0.05), education years, and English performance score (ANOVA, *p* > 0.05).

### 3.2. Clustering-Based Analysis

Figure 2A illustrates the spatial distribution of the twelve functional networks on the brain surface. Two recurrent transient connectivity states were identified for the three groups (Figure 2B). State 1 showed a higher frequency of occurrence (frequency of occurrence = 60.28%, connective strength = 0.395 ± 0.199) and stronger connective strength than state 2 (frequency of occurrence = 39.72%, connective strength = 0.174 ± 0.363). The number of transitions between the two states of the LB group was significantly higher than monolinguals (ANCOVA, FDR corrected q = 0.048), whereas comparable transition probabilities between states were observed in EB and monolingual cohorts (ANCOVA, FDR corrected q > 0.05). In addition, fraction time, dwell time, and connection strength showed no statistical differences between the three groups (ANCOVA, FDR corrected q > 0.05).

### 3.3. Dynamic Laterality of the Cortex

Among the three dynamic literality metrics (i.e., mean DLI, LR, and LF), only LR showed significant between-group differences (Figure 3). Specifically, the LBs showed reduced LR of the left lateral occipital cortex (L_LO2) and right prefrontal cortex (R_9p) relative to the EB group (ANCOVA, FDR corrected q < 0.05). Compared with the EBs, monolinguals showed significantly decreased LR mainly located in the left medial prefrontal cortex (L_10r), left lateral temporal cortex (L_PHT), L_LO2, and right inferior parietal cortex (R_IP1) (ANCOVA, FDR corrected q < 0.05). No statistical differences of LR were found between the LB and monolingual cohorts (FDR corrected q > 0.05). The comparison details are given in Table 2.

### 3.4. Correlation between Dynamic Measures and L2 Proficiency

Fifty-five participants who had L2 performance scores in the three groups were combined (including 20 EBs, 24 LBs, and 11 monolinguals), and Spearman’s ρ was calculated between the L2 performance score and the three regional dynamic laterality measures (i.e., mean DLI, LR, and LF) of each brain region, respectively. Furthermore, we also computed the two-state network dFC with the L2 performance score of Spearman’s ρ. We found that the LR of the left anterior superior temporal cortex (L_STGa)—a key region of the LAN—negatively correlated with L2 proficiency (ρ = −0.48, *p* < 0.0005, Figure 4), and dFC between CON and DAN in state 2 negatively correlated with L2 proficiency (Spearman’s ρ = −0.45, *p* < 0.0005). Moreover, correlations between L2 performance scores and the other two laterality measures (DLI and LF) and dynamic metrics (fraction time, mean dwell time, and transitions) were not statistically significant across the cortex. In addition, L1 proficiency did not significantly correlate with the three dynamic laterality measures.

## 4. Discussion

This article explored how being bilingual affects the way that the brain dynamically communicates between brain regions and functional networks. By dividing bilingual participants into those who learned their L2 early (EB) and those who learned it later (LB), distinct patterns of dFC were observed. The LB group showed more active state switching than monolinguals, while the EB group demonstrated higher levels of dynamics laterality reversal (LR) in specific areas of the DMN, V2, and DAN than both the LB and monolingual groups. Additionally, changes in LR in the left anterior STG were able to predict L2 performance. These results suggest that the way bilingual experience shapes dynamic brain connectivity is dependent on the age at which the L2 was acquired.

Generally, a higher frequency of transitions between different dFC states indicates greater flexibility in brain functional [38], which is positively linked to cognitive performance [39]. The LB group had more frequent transitions between dFC states than monolinguals, suggesting that bilingual acquisition during adolescence can enhance the functional flexibility of the brain and lead to better cognition in bilingual population [40]. However, there was no statistically significant difference in the frequency of state transitions between the EB and monolingual groups, indicating that the acquisition of a L2 affects the dFC of the brain in an age-dependent manner. This aligns with a previous study indicating no impact of learning L2 in early childhood on cortex development, but L2 acquisition after achieving proficiency in the first language can alter the brain structure [41].

LR reflects the number of times the functional activity between left and right brain associations reverses, as indicated by the zero crossings of laterality in two consecutive windows [34]. Higher LR suggests more frequent interhemispheric interactions. Interestingly, we found that the LR of several bilingual-related brain regions decreased in the order of EB, LB, and monolingual, suggesting that the earlier the individuals acquire L2, the more frequent their interhemispheric interactions may be. A previous study suggested that the left brain is associated with literal meaning comprehension (left laterality), while the right homotopic areas are involved in dealing with difficult metaphors [42]. Therefore, bilingual language processing, such as switching between languages, may require frequent reversal of functions dominated by different hemispheres. The brain regions that showed evident LR changes were mainly located in the left lateral occipital cortex, lateral and medial prefrontal cortex, left posterolateral temporal cortex, and right inferior parietal gyrus, which belong to V2, DMN, DAN, and FPN, respectively. These regions are deeply involved in language processing, such as word processing during reading [43,44], lexicosemantic processing [45,46], sensory-visceromotor link concerned with social behavior, mood control [47], motivational drive, directing visual attention, and short-term memory processes [48,49]. In bilinguals, these brain regions have been shown to have stronger intrinsic FC in DMN and FPN relative to monolinguals [16]. Furthermore, Wu et al. have indicated that moderate changes in dynamic laterality correlate with better cognitive performance [16], suggesting that our findings might indirectly support better cognitive abilities and less cognitive decline in the bilingual population [50,51].

We also found that the LR of the left anterior STG was negatively correlated with L2 performance. This result is consistent with previous literature indicating that LR and language task difficulty are negatively correlated [34]. The left anterior temporal cortex is a critical component of the language network [52,53] that integrates multiple words into more complex meanings [53,54]. Despite evidence suggesting that this region is not sensitive to whether the received words are from the same or different languages [55], the increased LR may suggest frequent shifts from more ipsilateral to highly contralateral connectivity, which may potentially impair language functions specific to the left brain and leading to reduced L2 performance. Meanwhile, we also found that the FC between CON and DAN in state 2 negatively correlated with L2 proficiency, indicating that the more skilled L2 usage was associated with weaker dynamic interactions between CON and DAN, especially in the weak connective state. Since then, DAN and CON have been indicated to be involved in language processing and word comprehension, respectively [56]. This result may suggest that two networks inhibit each other when processing different languages in a weakly connected state.

In the present study, no significant differences were found in the dFC of the two connectivity states between each pair of the three groups. However, previous literature has reported altered FC in bilingual population compared with monolinguals in the resting state using static FC analysis [16,18,21]. We speculated that such inconsistency may be attributed to methodological confounds, e.g., the static FC analysis focuses on the alterations in connectivity strength, whereas the dynamic FC analysis captures variations in connectivity strength over time. The inconsistency, we speculated, may be attributed to language and methodological confounds, such as variations in language families, sample size, acquisition protocols, analytic method (e.g., dFC analytic methods), definition of nodes and edges, etc., indicating the necessity of a validation study on a large sample set using a consistent analysis pipeline [57]. Therefore, our results were not in conflict with previous findings derived from static FC analysis but indicated insignificant association between bilingual experience and dFC.

There are several limitations that can be addressed through future work. First, the sample size of this study was limited, and there were missing demographic details for some of the participants. Replication on a larger, independent dataset may help validating our findings. Second, there may be unknown influences to the results due to ethnicity bias, although we have controlled for this effect during statistical analysis. Furthermore, some potential factors, such as IQ, socio-economic status, and difference in the languages, may bring unknown effects to the analysis. However, since these information was not provided by the dataset, the influences of these factors should be examined in future work. Given that the majority of the enrolled participants (>80%) were full-time university students, it is reasonable to assume that their IQ and basic cognitive functioning were within the normal range. Furthermore, the lack of the L2 acquisition age of each participant limited the direct estimation of dFC changes with age of L2 acquisition, which can be an important way to examine the reliability of this work. In addition, the TR (= 875 ms) of the data was shorter than conventional fMRI sequence (TR = 2000–3000 ms). This was a strength in the sense that it provided better temporal resolution. However, it can also increase the number of time windows, thereby making it more challenging to find optimized clustering centers. Analytic methods for fMRI with short TR were needed to address this issue.

## 5. Conclusions

In conclusion, our study showed that earlier L2 acquisition might induce more frequent interactions between bilateral brain regions related to language processing and short-term memory processing, and learning L2 in adolescence increases dynamics of functional networks. Furthermore, the dynamic laterality of activity in the left anterior STG can predict L2 proficiency. These findings can enrich our understanding of how bilingual acquisition alters brain functional networks and may also help elucidating the connectomic mechanisms of bilingualism.

## Figures and Tables

**Figure 1 brainsci-13-01231-f001:**
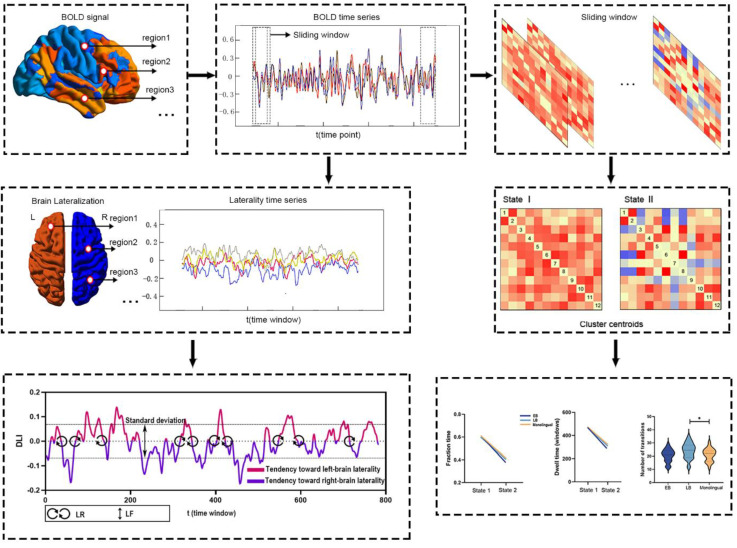
Analytic pipeline in the present study. Laterality of brain activity (i.e., DLI, LR, and LF) was calculated based on the initial time series (Red and blue represent positive and negative BOLD signals, red represents positive values and blue represents negative values). The time series of brain regions was segmented into continuous time windows, and the functional connectivity matrix at each window was constructed. The k-means clustering approach was used to identify the optimal dFC states of the three groups, and the dynamic metrics of dFC (i.e., fraction time, dwell time, and number of transitions) were then computed (* represents a significant difference between the two groups).

**Figure 2 brainsci-13-01231-f002:**
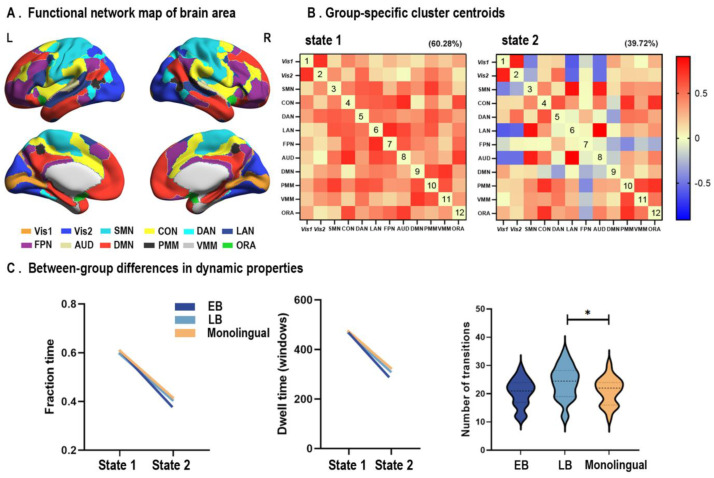
Comparison of the dFC state metrics among the three groups. (**A**) Mapping of the 12 functional networks on the cerebral cortex. (**B**) The identified two dFC states across the three groups. Values in the matrix represent Fisher’s z-transformed Pearson correlation coefficient. (**C**) Between-group comparisons of the dynamic metrics. The number of state transitions in LBs was significantly higher than that of the monolingual group (ANCOVA, FDR corrected, q < 0.05). Fraction time and dwell time of the two states did not show statistical differences (* represents a significant difference between the two groups).

**Figure 3 brainsci-13-01231-f003:**
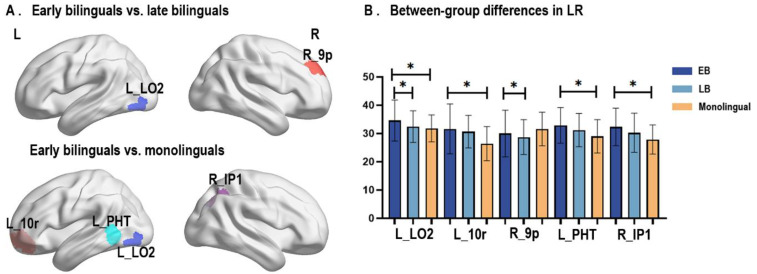
Between-group comparison of LR across the cortex. (**A**) Visualization of brain regions with statistic differences in EB vs. LB and EB vs. monolingual. (**B**) Bar plot of LR changes in these brain regions in the three groups. * q < 0.05 after FDR correction. Colors indicate brain regions belong to the functional network in the same color as shown in Figure 2.

**Figure 4 brainsci-13-01231-f004:**
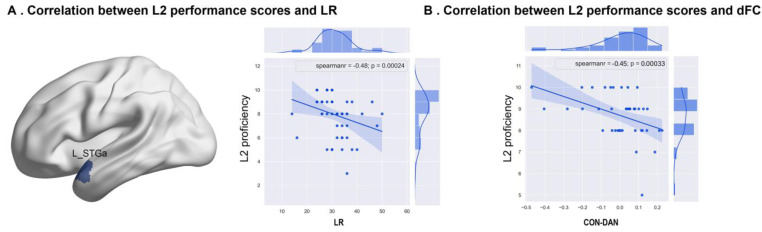
Correlation between L2 performance scores and LR and dFC. (**A**) LR of the left anterior superior temporal cortex (L_STGa) significantly correlated with L2 proficiency (Spearman’s ρ = −0.48, *p* < 0.0005). (**B**) dFC between CON and DAN in state2 significantly correlated with L2 proficiency (Spearman’s ρ = −0.45, *p* < 0.0005).

**Table 1 brainsci-13-01231-t001:** Demographic and clinical information of participants.

Subject	EB	LB	Monolinguals	*p*-Value
Age (years)	22.56 ± 0.50	22.03 ± 0.44	20.67 ± 0.43	<0.05 ^a^
Sex (women/men)	16/7	14/16	18/13	n.s. ^b^
Education years	14.61 ± 0.22	14.53 ± 0.19	14.13 ± 0.19	n.s. ^a^
Ethnicity (Hispanic or Latino/a; Hispanic or Latino/a and White; White)	15/6/2	1/0/29	0/0/31	<0.0001 ^a^
Spanish performance score	9.10 ± 0.19	8.15 ± 0.17	5.76 ± 0.16	<0.0001 ^a^
English performance score	9.74 ± 0.09	9.87 ± 0.08	9.77 ± 0.08	n.s. ^a^

Abbreviations: EB, early bilingual; LB, late bilingual; n.s., non-significant. Note: values are mean ± SD. ^a^ ANOVA. ^b^ Chi-square test.

**Table 2 brainsci-13-01231-t002:** Differences of LR between the three groups.

Label in MMP Atlas	Abbreviation	Location	Networks	Coordinates	Corrected *p*-Value
**EB vs. LB**					
42	L_LO2	Left lateral occipital cortex	V2-L	−49, −76, −10	0.0272
141	R_9p	Right prefrontal cortex	DMN-R	14, 51, 37	0.0272
**EB vs. MC**					
274	L_PHT	Left lateral temporal cortex	DAN-L	−55, −59, −16	0.0272
130	L_10r	Left medial prefrontal cortex	DMN-L	−6, 50, −3	0.0272
42	L_LO2	Left lateral occipital cortex	V2-L	−49, −76, −10	0.0119
289	R_IP1	Right inferior parietal cortex	FPN-R	33, −66, 45	0.0272

DMN, default mode network; V2, secondary visual network; DAN, dorsal attention network; FPN, frontoparietal network; L, left; R, right.

## Data Availability

Imaging data were downloaded from the OpenNeuro database (https://openneuro.org/datasets/ds001747).

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
