# Peer review of "Age of Bilingual Onset Shapes the Dynamics of Functional Connectivity and Laterality in the Resting-State"

_brainsci, 2023, doi:10.3390/brainsci13091231_

Round 1

Reviewer 1 Report

Overall this manuscript contributes nicely to the bilingualism literature.  Most existing studies applying resting state functional connectivity draw conclusions about static BOLD signal changes within the brain rather than time-dependent alterations over a longer window of time.  Authors recruited 3 groups: Early bilinguals, late bilinguals, and monolinguals.  The participants differed not only with regard to their age of L2 acquisition, but also with regard to their L2 proficiency.  The paper is well-crafted and easy to read.  However, a few clarifications/further analyses would be helpful to strengthen the quality of the paper.

  1.  How were early and late bilingual definitions determined? In other words, why isn't a bilingual who acquired an L2 at age 10 years considered an early rather than a late learner given what we know about sensitive periods in development?   2.  How were Spanish and English performance scores calculated? What did these tests assess (eg grammar, phonology, vocabulary)?  This should be explained within the participant section of the Methods.   3.  How do you think the proficiency differences, rather than age of L2 acquisition, impacts the results? Consider an analysis looking at the effects of proficiency on dFC beyond that performed for laterality reversal.   4.  Could an analysis with age of acquisition as a continuous variable be performed rather than only grouping participants together arbitrarily?  

Author Response

Response to Reviewers

We thank the Review for the insightful comments. We have provided point-to-point responses for the major concerns.

Reviewer 1

Overall this manuscript contributes nicely to the bilingualism literature.  Most existing studies applying resting state functional connectivity draw conclusions about static BOLD signal changes within the brain rather than time-dependent alterations over a longer window of time.  Authors recruited 3 groups: Early bilinguals, late bilinguals, and monolinguals.  The participants differed not only with regard to their age of L2 acquisition, but also with regard to their L2 proficiency.  The paper is well-crafted and easy to read.  However, a few clarifications/further analyses would be helpful to strengthen the quality of the paper.

Major issues:

  1. How were early and late bilingual definitions determined? In other words, why isn't a bilingual who acquired an L2 at age 10 years considered an early rather than a late learner given what we know about sensitive periods in development?

Response 1.1: Thank you for bringing up these questions for us to give a better explanation. Existing evidence have suggested that the "critical period" for learning a language as a native speaker may end around adolescence [1] and that the window for acquiring a native accent is between the ages of 6 and 10 [2]. Therefore, this study used an age range that attempted to honor developmental differences by defining early bilinguals as those who began to use a second language before 10 years old and late bilinguals as those who began to use a second language after the age of 14. Similar definition of EB and LB can also be found in [3].

1.2 How were Spanish and English performance scores calculated? What did these tests assess (eg grammar, phonology, vocabulary)? This should be explained within the participant section of the Methods.

Response 1.2: Thanks for the suggestion. We have added details of the specific methods of scoring in English and Spanish on page 3, line 21.

“The participants' English proficiency and Spanish proficiency tests were designed as follows: the study design called for all participants completed an English proficiency test, and any participant with experience with the Spanish language (from taking a semester in junior high to growing up with the language) to take a Spanish proficiency test. The tests were provided by Emmersion Learning, who administered the exams via an online platform, with participants completing them from a home computer. Both proficiency tests used elicited imitation (EI), in which the test-taker hears an utterance in the target language and is prompted to repeat the utterance as accurately as possible. If the participant is completely unfamiliar with the language of the utterance they hear, each syllable of the utterance will count towards that limited capacity, reducing their ability to accurately repeat the utterance to only a few syllables. In this way, EI can reliably approximate a learner’s proficiency level by measuring the accuracy of the repetition of utterances of increasing length and complexity[4].”

1.3 How do you think the proficiency differences, rather than age of L2 acquisition, impacts the results? Consider an analysis looking at the effects of proficiency on dFC beyond that performed for laterality reversal.

Response 1.3: Thanks for the suggestion. We reanalyzed proficiency as a covariate, and the results of the differences remained. And we have analyzed the effect of proficiency on dFC. The new results are shown in Figure 4, please see below. And changes were made to section 3.4 on page 8.

3.4. Correlation between dynamic measures and L2 proficiency

Fifty-five participants who had L2 performance scores in the three groups were combined (including 20 EBs, 24 LBs, and 11 monolinguals), and Spearman’s ρ were calculated between L2 performance score and the three regional dynamic laterality measures (i.e., mean DLI, LR, and LF) of each brain region, respectively. And we also computed the two-state network dFC with L2 performance score of Spearman’s ρ. We found that LR of the left anterior superior temporal cortex (L_STGa)—a key region of the LAN—negatively correlated with L2 proficiency (ρ = -0.48, p < 0.0005, Figure 4) and dFC between CON and DAN in state2 negatively correlated with L2 proficiency (Spearman’s ρ = -0.45, p < 0.0005). And correlations between L2 performance scores and the other two laterality measures (DLI and LF) and dynamic metrics (fraction time, mean dwell time and transitions) were not statistically significant across the cortex. In addition, L1 proficiency did not significantly correlate with the three dynamic laterality measures.”

And the results are discussed in the discussion section on page 10, line 19.

 “Meanwhile, we also found that the FC between CON and DAN in the state 2 negatively correlated with L2 proficiency, indicating the more skilled L2 usage was associated with weaker dynamic interactions between CON and DAN, especially in the weak connective state. Since the DAN and CON have been indicated to be involved in language processing and word comprehension, respectively [5]. This result may suggest that two networks inhibit each other when processing different languages in a weakly connected state”

Figure 4. Correlation between L2 performance scores and LR and dFC. (A) LR of the left anterior superior temporal cortex (L_STGa) significantly correlated with L2 proficiency (Spearman’s ρ = -0.48, p < 0.0005). (B) dFC between CON and DAN in state2 significantly correlated with L2 proficiency (Spearman’s ρ = -0.45, p < 0.0005).

1.4 Could an analysis with age of acquisition as a continuous variable be performed rather than only grouping participants together arbitrarily?  

Response 1.4: We agree with the reviewer that a correlation analysis may provide supportive evidence to our findings. However, the data we used are from a public dataset, which does not provide the acquisition age of the L2 of each subject. We have added this as a limitation in the limitation section, please see below.

“The lack of the L2 acquisition age of each participant limited the direct estimation of dFC changes with age of L2 acquisition, which could be an important way to examine the reliability of this work.”

Reference

  1. Lightbown, P.M., et al., How languages are learned. Vol. 2. 1999: Oxford university press Oxford.
  2. Thompson, I.J.L.l., Foreign accents revisited: The English pronunciation of Russian immigrants. 1991. 41(2): p. 177-204.
  3. Archila-Suerte, P., et al., The effect of age of acquisition, socioeducational status, and proficiency on the neural processing of second language speech sounds. 2015. 141: p. 35-49.
  4. Gold, C.E., Exploring the Resting State Neural Activity of Monolinguals and Late and Early Bilinguals. 2018, Brigham Young University: Brigham Young University.
  5. Vaden, K.I., et al., The cingulo-opercular network provides word-recognition benefit. 2013. 33(48): p. 18979-18986.

Reviewer 2 Report

The study examined how the age of bilingual onset shaped the dynamics of functional connectivity and laterality in the resting-state by analyzing publicly available fMRI images and demographic data. The results revealed that age of bilingual onset modulated the areas involved in visual language processing as well as such executive control as attention and working memory related areas.  I found the manuscript compelling and ready for publication in Brain Sciences journal upon addressing of the two minor points I list below.

Please add references 54 and 55  - they are currently missing. 

Author Response

Response to Reviewers

We thank the Review for the insightful comments. We have provided point-to-point responses for the major concerns.

Reviewer 2

The study examined how the age of bilingual onset shaped the dynamics of functional connectivity and laterality in the resting-state by analyzing publicly available fMRI images and demographic data. The results revealed that age of bilingual onset modulated the areas involved in visual language processing as well as such executive control as attention and working memory related areas.  I found the manuscript compelling and ready for publication in Brain Sciences journal upon addressing of the two minor points I list below.

Major issues:

  1. Please add references 54 and 55 - they are currently missing. 

Response 2.1: Thanks for pointing out this issue. We have added the missing references to the main text.

Reviewer 3 Report

This is an interesting study on age of bilingual onset and the dynamics of functional connectivity and laterality in the resting-state. The study is well-conducted and I agree that it may contribute to the literature. I have several comments to improve the manuscript further:

1. There could be better transitions between ideas, especially between paragraphs. The switch from discussing neuroimaging findings to dynamic functional connectivity comes abruptly, and might benefit from smoother transitions.

2. In the earlier part of the first paragraph, there should be more elaboration and comprehensive review of existing literature on the cognitive benefits of L2 acquisition age. The papers that the authors cited are mostly about bilingualism in general, not specific age of L2 acquisition. Given that the focus on the current study is on age of bilingual onset, the introduction seems abrupt and doesn't have good transition. I would suggest the authors to elaborate more studies on age of L2 acquisition and cognitive functions. Some relevant example:

Does early active bilingualism enhance inhibitory control and monitoring? A propensity-matching analysis. (2019). Journal of Experimental Psychology: Learning, Memory, and Cognition, 45(2), 360–378.

 Is there a relation between onset age of bilingualism and enhancement of cognitive control?.(2011). Bilingualism: Language and cognition, 14(4), 588-595.  

3. In the method section, the authors should consider elaborate more on the language characteristics of their sample. This issue has been criticized in many bilingual advantages literature

Surrain, S., & Luk, G. (2019). Describing bilinguals: A systematic review of labels and descriptions used in the literature between 2005–2015. Bilingualism: Language and Cognition, 22(2), 401-415.   4. The manuscript mentioned the OpenNeuro database and a specific dataset (https://openneuro.org/datasets/ds001747). It will be great for the authors to provide a brief description of this database and why it was chosen for your study. Additionally, clarify whether your sample is a subset of a larger dataset. If it is, provide the reasons for choosing this specific subset.   5. The selection criteria for the MC group is slightly not clear. Are these participants monolingual English speakers who've never been exposed to any other languages? It is important to clarify how the groups are determined   6. Potentially influential factors such as IQ, socio-economic status, or even the specific languages being learned were not accounted for. These factors could significantly affect brain connectivity and language learning. There should be discussion on this issue.     7. The article states that there was no significant difference in dFC between the bilingual and monolingual groups. This appears to contradict previous studies. It would be beneficial if the authors could explore this in more depth, possibly discussing why their results might have differed.

Author Response

Response to Reviewers

We thank the Review for the insightful comments. We have provided point-to-point responses for the major concerns.

Reviewer 3

This is an interesting study on age of bilingual onset and the dynamics of functional connectivity and laterality in the resting-state. The study is well-conducted and I agree that it may contribute to the literature. I have several comments to improve the manuscript further:

Major issues:

  1. There could be better transitions between ideas, especially between paragraphs. The switch from discussing neuroimaging findings to dynamic functional connectivity comes abruptly, and might benefit from smoother transitions.

Response 3.1: Thanks for the suggestion. We have added the following sentence to smooth the transition between paragraphs in Introduction section on page 2, line 16.

“In the neuroimaging studies, FC is an important indicator measuring the coactivation pattern between distributed brain regions.”

  1.  In the earlier part of the first paragraph, there should be more elaboration and comprehensive review of existing literature on the cognitive benefits of L2 acquisition age. The papers that the authors cited are mostly about bilingualism in general, not specific age of L2 acquisition. Given that the focus on the current study is on age of bilingual onset, the introduction seems abrupt and doesn't have good transition. I would suggest the authors to elaborate more studies on age of L2 acquisition and cognitive functions. Some relevant example:

Does early active bilingualism enhance inhibitory control and monitoring? A propensity-matching analysis. (2019). Journal of Experimental Psychology: Learning, Memory, and Cognition, 45(2), 360–378.

 Is there a relation between onset age of bilingualism and enhancement of cognitive control? (2011). Bilingualism: Language and cognition, 14(4), 588-595. 

Response 3.2: Thanks for helping us broaden the literature. We have included these studies in the first paragraph of the Introduction, line 10, as shown below.

“In addition, the early learning of a second language could modulate the development of cognitive functions (e.g., stronger inhibitory control [1], higher language proficiency, and better cognitive control [2]) as well as the brain structure (e.g., broader area of the right parietal suprachiasmatic cortex [3]).”

  1. In the method section, the authors should consider elaborate more on the language characteristics of their sample. This issue has been criticized in many bilingual advantages literature.

Surrain, S., & Luk, G. (2019). Describing bilinguals: A systematic review of labels and descriptions used in the literature between 2005–2015. Bilingualism: Language and Cognition, 22(2), 401-415.

Response 3.3: Thanks for the suggestion. We have elaborated the descriptions regarding the linguistic characteristics of participants and the method for assessing language proficiency on page 3, lines 9 and 21, as shown below.

“Individuals were invited to participate based on their survey responses, specifically whether they would fit into one of the three research groups. The EB group was composed of individuals proficient in English and Spanish with the experience of both languages began before the age of ten. The LB group consisted of individuals proficient in English and Spanish and who started to learn the L2 (Spanish) after fourteen years old. The MC group included individuals who only had English experience or had experiences with other languages (in addition to English) which were acquired at any age but was self-rated as “Novice” level [4].”

“The participants' English proficiency and Spanish proficiency tests were designed as follows: the study design called for all participants completed an English proficiency test, and any participant with experience with the Spanish language (from taking a semester in junior high to growing up with the language) to take a Spanish proficiency test. The tests were provided by Emmersion Learning, who administered the exams via an online platform, with participants completing them from a home computer. Both proficiency tests used elicited imitation (EI), in which the test-taker hears an utterance in the target language and is prompted to repeat the utterance as accurately as possible. If the participant is completely unfamiliar with the language of the utterance they hear, each syllable of the utterance will count towards that limited capacity, reducing their ability to accurately repeat the utterance to only a few syllables. In this way, EI can reliably approximate a learner’s proficiency level by measuring the accuracy of the repetition of utterances of increasing length and complexity [4].”

  1. The manuscript mentioned the OpenNeuro database and a specific dataset (https://openneuro.org/datasets/ds001747). It will be great for the authors to provide a brief description of this database and why it was chosen for your study. Additionally, clarify whether your sample is a subset of a larger dataset. If it is, provide the reasons for choosing this specific subset.

Response 3.4: Thanks for the suggestion. We have included a brief description of this dataset on page 3, line 2. In addition, the dataset we used is not a subset of a larger dataset.

“This dataset was chosen because of its well designed data collection, e.g., categorization of three groups based on L2 acquisition age, and strictly controlled for age, gender, and education (over 80% participants were full-time college students), which could minimize the influences of these factors on statistical analysis.”

  1. The selection criteria for the MC group is slightly not clear. Are these participants monolingual English speakers who've never been exposed to any other languages? It is important to clarify how the groups are determined.

Response 3.5: The MC group included individuals who only had English experience or had experiences with other languages (in addition to English) which were acquired at any age but was self-rated as “Novice” level. Please refer to Response 3.3.

  1. Potentially influential factors such as IQ, socio-economic status, or even the specific languages being learned were not accounted for. These factors could significantly affect brain connectivity and language learning. There should be discussion on this issue.

Response 3.6: Thanks for the suggestion. Most of the participants included in this study were full-time college students, which could, to some extent, reduce the influence of education and IQ. We agree with the reviewer that these factors may have potential influences on the results. However, because the dataset does not provide these information, we have added the following sentence to the Limitation on page 10, line 23.

“Some potential factors, such as IQ, socio-economic status, and difference in the languages, may bring unknown effects to the analysis. However, since these information was not provided by the dataset, the influences of these factors should be examined in the future work.”

  1. The article states that there was no significant difference in dFC between the bilingual and monolingual groups. This appears to contradict previous studies. It would be beneficial if the authors could explore this in more depth, possibly discussing why their results might have differed.

Response 3.7: Please allow us to clarify. There are few previous studies on bilingual dFC, and only one on Cantonese, which is different from the language family of the present study and may be the main reason for the difference in results. In addition, other reasons may also contribute, such as analysis methods, data acquisition strategies, node definitions. And we've added this part of the explanation to the discussion section.

“The inconsistency, we speculated, may be attributed to language and methodological confounds, such as variations in language families, sample size, acquisition protocols, analytic method (e.g. dFC analytic methods), definition of nodes and edges, etc., indicating the necessity of a validation study on a large sample set using a consistent analysis pipeline [5].”

Reference

  1. Hartanto, A., H.J.J.o.E.P.L. Yang, Memory,, and Cognition, Does early active bilingualism enhance inhibitory control and monitoring? A propensity-matching analysis. 2019. 45(2): p. 360.
  2. Luk, G., et al., Is there a relation between onset age of bilingualism and enhancement of cognitive control? 2011. 14(4): p. 588-595.
  3. Wei, M., et al., How age of acquisition influences brain architecture in bilinguals. 2015. 36: p. 35-55.
  4. Gold, C.E., Exploring the Resting State Neural Activity of Monolinguals and Late and Early Bilinguals. 2018, Brigham Young University: Brigham Young University.
  5. Zheng, W., et al., Preterm‐birth alters the development of nodal clustering and neural connection pattern in brain structural network at term‐equivalent age. 2023.

Round 2

Reviewer 3 Report

All my comments were addressed well by the authors. I appreciate their hard work.